# Parents’ Perspectives on the Health Education Provided by Clinicians in Portuguese Pediatric Hospitals and Primary Care for Children Aged 1 to 10 Years

**DOI:** 10.3390/ijerph17186854

**Published:** 2020-09-19

**Authors:** Anabela Pereira, Joaquim Escola, Vitor Rodrigues, Carlos Almeida

**Affiliations:** 1Northeast Local Health Unit, Mirandela Hospital Unit, 5370-210 Mirandela, Portugal; anabelapereira83@gmail.com; 2Institute of Philosophy of the University of Porto, School of Human and Social Sciences, University of Trás-os-Montes e Alto Douro, 5000-801 Vila Real, Portugal; jescola@utad.pt; 3Research Center in Sports Sciences, Health Sciences and Human Development, CIDESD, 5000-801 Vila Real, Portugal; calmeida@utad.pt; 4School of Health, University of Trás-os-Montes e Alto Douro, 5000-801 Vila Real, Portugal

**Keywords:** health promotion, child, parents, nursing

## Abstract

This study aimed to analyze parents’ perspectives of the health education practice provided by health professionals to children and parents. This is a descriptive research with a qualitative approach, based on the conceptual framework of health education provided by health professionals to children/young people and families. The selection took place by non-probabilistic sampling of convenience, and was developed with parents/users of health units for pediatric hospitalization and primary health care in northern Portugal. Data were collected using semi-structured interviews with 20 parents from March to April 2019, and were analyzed using the content analysis technique. Health education was understood to enhance health, and the evaluation was carried out according to planning and partnership. The most frequent topics were food, the national vaccination plan, and accident prevention. The evaluation shown is very positive, pointing to the nurse as the professional of choice for this practice and thus contributing to the visibility of nursing. We can state that the perspectives of parents about the health education carried out by health professionals show a practice that values health; is adaptable to the complexity of the binomial child and parents; is capable of influencing health determinants with sustainability, efficacy, and effectiveness; and gives visibility to nurses’ positions as health educators.

## 1. Introduction

The health of children and adolescents, over the last century, has experienced an epidemiological transition worldwide, namely due to the increase in chronic disease in the child population [1]. Likewise, the care provided by parents during childhood can also affect the child’s development, because of the existence of individual and social factors that are faced by the family, and can be a challenge [2]. Therefore, it is understood that the evidence indicates that investment in quality programs in early childhood has a high rate of return for society [3].

Considering the importance of knowing how to manage disease, informed decisions, and healthy behaviors, health professionals are therefore needed who are capable of promoting the development of self-confidence and self-care in children/young people and the family [4]. Skills in Health Promotion (HP) have also been discussed in international contexts, implying their improvement in a reflective practice [5].

### 1.1. Health Education Concept

If HP constitutes a global political and social process that encompasses actions directly aimed at strengthening the abilities and capacities of individuals that modify social, environmental, and economic conditions, in order to mitigate their impact on the public and on individual health, health education (HE) is considered as an educational process to promote and educate about factors that affect the general population and each individual, in particular teaching behaviors and motivating change to create healthy behaviors [6]. HE is a set of opportunities consciously built for learning involving some form of communication aimed at increasing literacy in health, including increasing knowledge and attitudes that are conducive to individual and community health [7].

### 1.2. Health Education in Portugal

The health status of Portuguese people has improved considerably over the last decade, although health inequalities are linked to a number of health determinants, including the living and working conditions of people, the physical environment, and a range of behavioral risk factors (smoking, alcohol use, diet, and physical inactivity). Bucking the normal trend for significant inequalities in this area, sharper declines in regular smoking have been seen for 15-year-old girls and boys, and these positive results are being reinforced by public health actions that target tobacco control. The prevalence of overweight and obesity among 15-year-olds remains close to the European Union (EU) average, and rates of physical inactivity for 15-year-olds are among the highest in EU countries, so Portugal has implemented national strategies for nutrition, preventing and treating obesity, and promoting physical activity [8].

However, Portugal has a shortage of health promotion and disease prevention activities, so in the next few years the strategies for public health include the reduction in risk factors for non-communicable diseases—in particular, the use of and exposure to tobacco/smoke and overweight and obesity in the school-age population—and a “Programme for Physical Activity” has also been put in place, which aims to promote healthy lifestyles and tackle sedentary behaviors.

For children from 0 to 6 years old there exists “Education Support Services” with an inter-sector responsibility, public/private partnership, an interdisciplinary approach, and family involvement [8]. It was defined as an “Early Childhood Intervention Framework”, based on an organizational system (“interservices”) and in collaboration with available resources. The focus was on encouraging the development of new projects and the adaptation of some others, thus increasing the number of children and families supported and practices oriented towards them. In turn, the ongoing progress is being developed regarding training (academic level and in-service training) [9]. There is also a “Young Health Programme” that encourages young people to take control of their health and well-being, build self-reliance and self-respect, and thereby avoid future risk behaviors; the focus for this is on the health and mental well-being of vulnerable pre-teens [10].

In this scenario, health professionals assume a strategic position due to their proximity to the child and their family [1], and thus educational interventions that support parents in the acquisition of positive parenting skills and creating safe and health-promoting environments are the goals of health professionals’ care [2]. In addition, with the intention of supporting practice with scientific evidence, professionals must demonstrate the quality and safety of their care through satisfactory patient results [11].

On the one hand, parents are responsible for caring, educating, loving, promoting autonomy, and preparing children for the challenges of the present and adult life; on the other hand, the literature does not clarify the role of health professionals in the development of positive parenting [12]. Thus, the assessment of those who are the target of care is of enormous importance, since it provides a picture of the effectiveness of health professional’s interventions and acts as an indicator of the quality of care in HE for child and parents.

Assuming that parents are partners, and also supervisors and evaluators of the health professional’s knowledge, skills, abilities, and technical and relational skills, it is important to know their perception of the concept of HE and the evaluation of practices in HE for children and parents.

In this sense, the present study has the general objective of analyzing the perspective of parents on the practice of HE performed by health professionals for children and parents. As a specific objective, we intended to identify the HE concept and the perception and evaluation of HE practices by parents.

## 2. Materials and Methods

This is a descriptive study with a qualitative approach, regarding the parents’ perspectives of the HE practices provided by health professionals to children and parents, and which is part of an investigation on the contribution of the HE practices provided by nurses to the health of the child and parents. The study was developed with parents/users of health units (pediatric hospitalization and primary health care) in northern Portugal.

In the first phase, an extensive literature review was carried out through bibliographic research related to the theoretical and social context of the phenomenon under study. Regarding the general objective, the appropriate descriptors for searching the databases were selected. The descriptors used were HE, HP, nurses, and children. In the inclusion criteria, we selected studies with various methodologies in the languages English, Spanish, or Portuguese, with up to 10 years of publication, and that specifically address the practice of HE in children/young people and parents performed by nurses. Articles that did not fit the objectives of the study or did not have relevant information for the study were excluded. The databases used were: Online Knowledge Library (B-On), EBSCOhost Online Research Databases (EBSCO), Psychology and Social Science Journals on the Web (PSYCLINE), Medical Literature Analysis and Retrieval System Online (MEDLINE), Scientific Electonic Library Online (SciELO), Elsevier, PubMed Central (PMC), and Portuguese Open Access Scientific Repository (RCAAP). The reading and analysis of relevant strategic documents was also carried out, including documents from international and national organizations.

In the data collection, semi-structured interviews were applied individually to parents, and this was conducted by the main investigator. The injury technique was chosen to obtain the perspective of the participant in order to achieve correspondence between theory and facts and to improve the understanding of the study’s problem. Thus, the interviews were prepared considering the theoretical framework of the study, according to the specific objectives of the study (to identify the HE concept and identify the perception and evaluation of HE practices by parents), and the professional experience of the researcher in HE for children/parents.

The interviews took place in primary health care (child health appointment) and in the hospital (pediatric service). The duration was about 30 min, and driven by a data collection instrument that contained 12 open questions (Table 1). The principal topics of the research were converted into colloquial language for the participants.

In a general way, the questions were divided in three sections: identifying the parent’s perception of the HE concept; identifying the HE practices performed by health professionals (frequency, location, frequent topics, professional of choice, HE planning, facilitator, difficult factors, and general objectives); and identifying the parent’s evaluation.

The selection of parents had the following inclusion criteria: attending a child health appointment in primary health (children aged 1 to 10 years) care within 1–31 March 2019; being present when the child is admitted to a pediatric hospital inpatient ward (children aged 1 to 10 years), within 1–30 April 2019. Regarding the sociodemographic characterization of the research participants, the following variables were included: age, gender, and interview place.

The option for a non-probabilistic sampling of convenience was due to the fact of having better accessibility to the population. The sample consisted of 20 parents.

For the analysis and interpretation of the data, the content analysis technique [13] was used, organized in three different chronological moments: pre-analysis; the exploration of the material; and the treatment of the results: inference and interpretation. In pre-analysis, the “floating reading” of the interviews allowed us to organize and systematize ideas, organize the material, know the context of the information, and highlight the generic guidelines. Groups of codes that expressed the same ideas were classified into categories. A data classification scheme was then structured according to the subcategories found, taking into account rules such as homogeneity, mutual exclusion, relevance, objectivity, fidelity, and productivity. The registration units were also coded. The interviews were recorded with a recording device, and later the audio was transcribed verbatim; in this process, we always took into account that the transcription of the interviews respected the content of the parents’ interviews so as not to skew the results and ensure the maintenance of ethical principles.

The names of the interviewees were also replaced by identification codes: PH1 to PH10 for the parents interviewed in primary health care (child health appointment), and P1 to P10 for the parents interviewed in the hospital (pediatric service).

The research was authorized by those responsible at the institutions that participated in the study, and we secured a favorable opinion from the respective ethics commissions of the Northern Regional Health Administration: Opinion n° 124/2018. The recommended ethical procedures were also followed. There was no relationship between the researcher and the study participants, which decreased the potential for the researcher to influence the study or for the study to influence the researcher. It was also explained to the interviewees that their participation would not entail risks, and that they had the full freedom to accept or leave the study without restrictions or consequences. They were guaranteed that there would be confidentiality regarding the collected data, which would be used exclusively for research. Privacy was also respected, and informed consent was obtained.

## 3. Results

Regarding the characterization of the research participants, the parents’ ages varied from 27 to 41 years, with the mean age of 33.5 years. All the participants were female. Regarding the interview place, 10 parents were interviewed in primary health care (child health appointment), and 10 parents were interviewed in hospital (pediatric service).

The systematization of the data extracted from the interviews was in line with the defined objectives, and reflected the perspective of each participant, as a result of their training, experience, and individual course.

According to the objectives and theoretical framework of the study, we pre-established eight categories (concept, frequent themes, HE planning, facilitating elements, difficult elements, HE general objectives, parent evaluation, and professional practice). We found subcategories, and the assigned names arose naturally from the participants’ words, and the concepts were part of this research conceptual framework.

From the collected data, it was then possible to identify 16 subcategories (Table 2).

### 3.1. Perceptions of Parents about the Health Education Concept

The interviews of the participants highlighted the HE concept as an important practice and facilitator of health literacy increase, which essentially deals with the transmission of knowledge, allowing the adoption of healthy lifestyles in the child/young person. Teaching and health habits were the most prominent responses, from 8 participants and 10 participants, respectively.

“That’s when they teach us to get habits to be healthy”.(PH5)

“It is a time when you learn how to adopt healthy lifestyles”.(P6)

All the participants also mentioned that nurses are the health professionals whom they choose first to clarify doubts related to the health of the child/young person.

“To the nurse because she is always present”.(P1)

“To the nurse because she is always here”.(P2)

“To the nurses”.(P3)

### 3.2. Assessing the Nurse’s Practices in Health Education to Child and Parents

The parents interviews expressed a HE practice performed frequently; it was found that in child/young person health appointment nurses always perform HE interventions, and in pediatric services it was found that HE was performed when the parents asked the nurses questions, or if the nurse thought it was appropriate.

“When I go to appointment they always teach”.(PH1)

“Whenever I ask, other times they are on their own initiative depending on the subject we are talking about at the moment”.(P5)

Regarding the most frequent themes approached by nurses during HE practice, the interviews raised three subcategories, “healthy lifestyles promotion”, “disease prevention”, and “child development”, verifying that there is a concern in addressing the priority thematic areas of HE. The subcategory “healthy lifestyles promotion” was the most mentioned (10 participants), highlighting healthy eating, vaccination, and accident prevention.

“Food, vaccines and accidents”.(PH3)

“In my case, we talk about taking the medication and healthy eating”.(P4)

In the subcategory “disease prevention” (seven participants), the concern with maintaining care for chronic illness and care after hospital discharge was highlighted, thus showing concern for a positive parental performance.

“It depends on what I want to know at the time, but it has always been related to my daughter’s disease”.(P3)

“In my case, I mainly talk about my son’s disease, which is Diabetes and that is why there is a lot of care to be taken”.(P2)

The subcategory “child development”, mentioned by three participants, pointed out the importance of parents understanding the stages of growth and the child/young person development.

“The child’s normal development”.(PH9)

“Prevention of accidents, vaccines and child’s normal development”.(PH8)

Regarding the nurse’s preparation of HE practice, the parents’ interviews pointed to two subcategories, “scripts/norms elaborated” and “improvise”, with the subcategory “scripts/norms elaborated” being the most mentioned (15 participants), showing the previous preparation of HE practice.

“Nurses are always prepared”.(PH3)

“Prepared because they had other parents with the same doubts as me”.(P5)

The subcategory “improvise” (mentioned by five participants) highlighted the fact that nurses take advantage of all opportunities to share their health knowledge, due to the various questions/doubts that may arise from parents.

“Improvised because they never know what we can ask”.(PH1)

“Improvised, because sometimes they even take advantage of it and speak to everyone in the infirmary”.(P6)

Regarding the elements that facilitate HE practice, the interviews of seven participants pointed to the subcategory “care partnership”, highlighting the importance attributed to the relationship of trust and proximity to the health professionals, and the parents’ consequent involvement and participation.

“Nurses demonstrate that they have health knowledge and help to clarify our doubts”.(PH1)

“Nurses help me a lot to take care of my son because they are here 24 h”.(P6)

Regarding the elements that are difficult in HE practice, four participants pointed to the health professionals’ limited time availability, and thus the subcategory “overload nurse’s work” emerged.

“The fact that sometimes it takes a long time to be attended”.(PH5)

“Yes. There are few nurses”.(P3)

“Yes, sometimes nurses don’t have much time to talk with us”.(P10)

Still, in relation to the difficult factors, three participants pointed to the difficulty in managing the scheduled work with the child/young person appointment schedule.

“Sometimes I have a hard time getting a layoff at my workplace”.(PH1)

“Yes, reconcile with working hours”.(P3)

### 3.3. Evaluation of Parents about the Health Education Practice

Regarding the category “HE general objectives”, the interviews pointed to four subcategories: “health literacy increase”, “healthy lifestyles promotion”, “disease prevention”, and “disease treatment”. The subcategory “healthy literacy increase” was the most mentioned (seven participants) showing the importance of acquiring health knowledge that promotes parents’ autonomy.

“Teach parents about everything related to children”.(PH9)

“To transmit knowledge and better prepare parents, especially first-time parents, as is my case”.(P7)

The subcategory “disease treatment” (mentioned by six participants) showed once again the concern for a positive parental performance.

“Teach how to deal with sick children”.(P9)

“Prepare parents and children so that we can do things well at home afterwards”.(P4)

The subcategory “disease prevention” (mentioned by four participants) reflected the importance of skill acquisition by parents to prevent disease in children.

“Prevent children from getting sick”.(PH5)

“Prepare to avoid the disease”.(P3)

Finally, the subcategory “health lifestyles promotion”, mentioned by three participants, again highlighted the importance attributed to health-promoting habits and behaviors.

“Make sure children are always healthy”.(PH4)

“Prepare for future situations”.(P2)

When the parents were asked about the HE evaluation developed by health professionals, the answers pointed to three subcategories: “positive evaluation”, “negative evaluation”, and “care partnership”. The subcategory “positive evaluation” was mentioned by 15 participants, showing their confidence in nurses’ knowledge.

“Nurses make good appointments”.(PH3)

“It makes us feel confident in taking care of my son”.(P9)

The subcategory “care partnership” (pointed out by four participants) demonstrated the importance attributed to the relationship of trust and closeness between health professionals and children and parents.

“It appears that they can also help us”.(PH1)

“Help them to learn with us too”.(P4)

One participant pointed to the lack of human resources, and consequently the nurses’ lack of time availability for HE practice.

“It helps them to better prepare appointment. It is a pity that they are few and sometimes take little time with us”.(PH5)

In the subcategory “nursing visibility/professional achievement”, 19 interviews showed a positive HE evaluation developed by nurses through the recognition of the importance of their professional activity.

“They do a good job because my son has rarely been sick thanks to what they have been teaching me”.(PH9)

“They show that they are good health professionals”.(P6)

“They show that they are a class of confidence”.(P9)

## 4. Discussion

The concept of HE envisaged by parents is based on an intervention focused on increasing health literacy, based on the transmission of health-promoting behaviors, which allows them to adopt healthy lifestyles, not focusing on maintaining disease.

However, in a study about nurses’ social representations, it was found that the education model aimed at individuals about disease was still guided by traditional education, and not at the centrality of these regarding their health situation [14]. In another study, HE was also understood as an information transmission tool [15]. Thus, this research shows a HE practice centered on health, and not only on disease treatment, which may indicate that health professionals, in general, begin to stop being connected to traditional pictures of limited and limiting HP practices, despite overwork, inadequate education, and practice, and focus on the biomedical model [16].

Despite this perspective, the importance given to this practice is shown in the present research, since all parents said that they did not miss children’s/young people’s health appointments, all demonstrated a positive evaluation of HE practice, and 16 participants reinforced the importance that HE has in professional nursing activity and its recognition. In fact, there is currently evidence of the increasing autonomy of nurses, with the expansion and consolidation of care and management actions and emancipatory practices of HP [17]. At the same time, HE is perceived as a frequently performed practice, and is evidenced taking advantage of opportunities, complying with current legal guidelines; the role of nurses, as professionals sensitive to emerging health needs and vulnerabilities, and the application of their knowledge in HP, is highlighted [17]. This result corroborates a study conducted in Ireland, in which parents yearned to participate actively in the educational process, together with health professionals, to implement care plans according to their children’s needs [18].

The nurse received a prominent role, when considered as the professional that parents turn to in the first place to clarify doubts, thus showing their humanized attention and focus on solving problems in the process of making decisions, and the bond of trust [19]. Indeed, several studies reported that trust is an element associated with high levels of parental satisfaction, the well-being of the child and parents, acceptance and adherence to treatment, and parental involvement in decision-making regarding children’s treatment [20].

Bearing in mind that education and family support are important in education for health care, through habits and lifestyles [21], the themes frequently addressed by nurses (mentioned by parents) showed their concern in addressing priority areas of HE, consequently complying with legal guidelines and having a positive influence on health determinants.

However, in one study, the results showed that there was a lot of work still to be done to promote healthy lifestyles, particularly among adolescents, because it is in this period of development that health behaviors are established [22]. However, we must take into account that the promotion of healthy behaviors in childhood is a determining factor for health, since positive changes have an impact on health outcomes [22]; also, in the literature, lifestyles are a priority theme in contemporary political agendas, namely in public health [23]. Thus, this result is a reflection of a practice that promotes children’s and parents’ health training and the adoption of behaviors that promote healthy lifestyles in order to reach the maximum health potential for the children/parents, contributing to the efficiency of public policies and health gains.

In the planning of HE interventions, we clarified a planning with the evaluation of health needs, and, consequently, lines of action capable of acting on health determinants, adapted to the complexity of both the children and parents, and each child’s/young person’s life cycle. In parallel, studies reported that, in order to intervene, planning based on evaluation is necessary, and thus effective implementation leads to well-planned activities [24], allowing health professionals to better take advantage of opportunities; better define objectives; organize activities and interventions; and monitor, inspect, and evaluate the results [25]. The use of improvisation demonstrated the capacity for creativity and taking advantage of all opportunities in HE practice.

In order to reflect good nursing practices, it is important to continuously search for the highest quality in care standards, which, in turn, depend on elements that act as the facilitators or hinderers of these standards [26]. The care partnership, pointed out in this research as a facilitating element of the HE performed by health professionals for children and parents, highlights the necessary relationship between health professionals and family, so that individuals make good choices and maintain healthy behaviors [4]. Consequently, this result shows the capacity and competence of mediation, negotiation, and building a trusting relationship on the part of health professionals, making parents feel like a fundamental part of the whole process. In fact, the care partnership is essential in caring for the child/young person and family, as the provision of a health professional’s care in the child area encompasses specific knowledge and skills to ensure quality care, and allows the child and family to feel like participants this in process of interaction and care [27]. Note that this relationship of the proximity, involvement, and participation of parents also promotes the ability to adopt responsible and conscious behaviors in health, and a practice directed to the health needs of the child and parents.

We also know that there are multiple challenging factors in the construction of good professional nursing practice [26], so the lack of time availability of nurses (pointed out by four participants) can indicate that nurses are overloaded with work, either due to the lack of human resources or due to the bureaucratic processes (records) inherent to nursing care, which can be a demotivating and impeding situation for the effective development of HE practice. Additionally, in a study the participants pointed out the difficulties observed in the performance of nurses in primary care, the work overload, and the problems of human resources [28]. In this sense, problems such as the lack of employees, the high number of users, or the exhaustion of work are often faced by health professionals [27], and have been frequently pointed out in nursing studies that aim to understand the lack of quality, adverse effects events, and professional dissatisfaction in health care [29].

Furthermore, also recognition is illustrated by the positive evaluation verified in this research, indicating that despite the existence of obstacles, health professionals remain committed to contributing to the effectiveness of interventions, which allows us to reach the maximum health potential of the child/parents and improves the visibility of professional practice. Thus, we can conclude that health professionals see the effective result of their work and the recognition of the care in the health of the child/parents, which, in turn, is an important motivating tool for better care.

Regarding the general objectives of HE from the parents’ perspective, they meet the HE concept definition, making the transmissive characteristic of knowledge that generates health-promoting styles, habits, and behaviors very clear, especially because HP advances giving priority to health and the construction of public policies and environments that support healthy choices [23].

In the interviews, the importance of acquiring parenting skills, which allow better adaptation to disease processes and to the management of an effective therapeutic regime, was also clear, and this is corroborated by other studies [30,31]. In a study about the experience of child chronic conditions, the importance of the family accumulating knowledge for the provision of care was found, as over time this is also learned and performed by the child [30]. In another study about the experience of transitions in parenting, it was found that nurses have an essential role in caring for parents who experience transitions in parenting [31]. It is concluded, therefore, that interventions in HP are an integrating path of care and are based on technical, scientific, and cultural knowledge, and have the capacity to produce individual and family changes. As HE is linked to HP, it is assumed that the family is empowered with autonomy and not just with disease prevention attitudes [32].

The importance attributed to autonomy in health maintenance, and the consequent acquisition of skills to prevent situations of illness was also evident, showing that the centrality of health professionals is established in the production of more autonomy, so that users and the family are allies in healthcare [26].

We know that a good nurse is built through their visibility in society, by their appropriate posture and respect, knowledge, quality of the work, and way of acting, and therefore the evaluation of their practice in addition to characterizing the profile of these professionals will directly reflect the effectiveness of nursing care [33]. In the present study, the parents’ evaluation was positive, which reflects the recognition of the nurse’s role as a health professional with the necessary knowledge to help in the care of child and parents, their active role, their interactive and interventional role in health decisions, and the effective transmission of knowledge that generates increased health literacy and health gains.

In the HE evaluation, parents emphasized again the care partnership, which naturally increases a sharing and co-responsibility intervention, based on a relationship of trust between health professionals and children and parents, distancing themselves from the mere transmission of information, and contributing to the reduction in vulnerabilities, favoring the health response capacity through the sharing of knowledge, obtaining child health indicators, and creating links between health and children’s rights [34]. In fact, parenting is a complex process, being one of the most challenging developmental transitions for the family, and as such it must be a focus of attention with high sensitivity to the provision of health care, in order to better provide knowledge and training to parents [31]. In Portugal, the governmental guidelines for “Early Childhood Intervention” are an integrated service provision that is child-centered and family-centered which includes preventive and habilitative interventions (education, health, and social ones) in order to facilitate child development and family–child interactions and enhance family skills [9].

Finally, the visibility of the professional activity of nursing in HE was also highlighted, recognizing the skills of the nurse as a health educator, which inherently leads to the improvement of the child/parent’s health. Bearing in mind that the HP referentials provide nursing education with a transformation of teaching practices, favoring the overcoming of the hegemonic model and technical rationality [28], in view of the result of this research, we can conclude that nurses exercise educational practices that have an important role. Additionally, in a study the role of nurses in HE was evident, with nurses being pointed out as the facilitator of these actions, the instigator of the team, and the articulator of this practice [15]. The literature also indicates that nurse’s interventions in different contexts use different educational approaches, breaking with the vertical transmission of information paradigm, generating participation and decision-making, thus highlighting their fundamental role in HP [35].

### Limitations of the Study

As a limitation of the study, the need for a larger group of participants can be pointed out, because with a small research sample it is difficult to generalize the results. The opinion of those who are the target of health care was important, as it brought evidence of recognition, and a positive assessment of the HE practice performed by health professionals, which allowed us to verify the effectiveness of care in child/young people and parents’ health. It would be interesting in future research to obtain a bigger sample.

## 5. Conclusions

This research identified HE practice performed by health professionals that was perceived by parents as adaptable to the complexity of the child–parents binomial, capable of providing proactive responses according to needs, influencing health determinants, enhancing protective health factors, and producing conscious and sustained changes.

The future challenge for the health system besides focusing on public participation and patient empowerment is also to be able to maintain the motivation of health professionals, particularly nurses, and including the implementation of effective measures to ensure financial sustainability, while improving underserved fields.

This research also shows the importance of providing training to health professionals to gain HP competences and consequently contribute to the sustainability, efficacy, and effectiveness of the HE practice performed by health professionals. That is, the data resulting from an experience shared by parents who participated in this study consolidated and reinforced the position of nurses as health educators, which consequently allowed the maintenance and improvement of the child/parent’s health, efficient health policies, the efficiency of health services, and gains in health.

## Figures and Tables

**Table 1 ijerph-17-06854-t001:** Semi-structured guide used with the participants.

1. Do you know what HE is?
2. When you take your child to a child health appointment, do the nurses do HE?
3. What themes do they address most?
4. Does it seem to you to be a prepared or improvised approach?
5. What are the objectives of the HE?
6. What is your evaluation of the HE performed by the nurses?
7. Do you think they reach the objectives?
8. Do you learn from nurses how to improve your child’s health?
9. Do you attend primary health care whenever you are called by a nurse?
10. When you have any questions about your child’s health, who do you ask?
11. Do you have factors that hinder you going to a child health appointment?
12. Do you consider that there are factors that hinder the HE performed by the nurses?

**Table 2 ijerph-17-06854-t002:** Presentation of the objectives, categories, and subcategories resulting from the parents’ interviews, Trás-os-Montes and Alto Douro, northern Portugal, Portugal, 2017 (*N* = 20).

Objectives	Categories	Subcategories
Identify the HE concept	Concept	Health literacy increase
Identify the HE practice to child and parents	Frequent themes	Healthy lifestyle promotion
		Disease prevention
		Child development
	HE planning	Scripts/norms elaborated
		Improvise
	Faciliting elements	Care partnership
	Difficult elements	Overload in nurse work
Identify the HE evaluation to child and parents	HE general objectives	Health literacy increase
		Healthy lifestyle promotion
		Disease prevention
		Disease treatment
	Family evaluation	Positive evaluation
		Negative evaluation
		Care partnership
	Professional practice	Nursing Visibility/professional achievement

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
