# Peer review of "Parents’ Perspectives on the Health Education Provided by Clinicians in Portuguese Pediatric Hospitals and Primary Care for Children Aged 1 to 10 Years"

_ijerph, 2020, doi:10.3390/ijerph17186854_

Round 1

Reviewer 1 Report

Dear authors. Thank you very much for involving me in your work. I enjoyed reading your paper as it covers a subject that really resonates with me. Below, I've outlined some considerations to take into account, with the sole purpose of trying to improve the original that, in my opinion, would be publishable after making just a few changes.

It seems to me that there's a big imbalance between the introduction (as a theoretical framework) and the discussion. Which, by the way, is for the benefit of the discussion, which I understand is very rich and attractive. What I'm trying to say is that it would be necessary to strengthen the theoretical framework of the study with further development based off scientific literature, as an introduction to its object of study.

In the introduction, as it is written right now, there are many generalisations at the global level. It's not that they're wrong, but I find there's a lack of a more vigorous approach to the object of study, as well as some sort of evaluation or comparison of a regional nature. I am sure there must be many regional differences around the world in the role of families and healthcare professionals when it comes to the Health Education that they can provide, as has been described. It would be very interesting to approach this subject somehow, even if it's only in order to position your study.

The research sample is very small, which makes it impossible to generalise the conclusions. Obviously you aren't going to restart the study right now with a larger sample size and, as it's presented, the scientific community would find the study very interesting, especially nursing professionals. So I'm simply mentioning this as an important limitation, as you've done in your manuscript.

In this sense, I also believe that it would be interesting to include a clearer explanation of the limitations, especially of its representativeness with respect to the general population under study. With reference to this point, I suggest that these limitations not be included in the discussion, but in a different section that also includes possible lines of future research arising from this study.

On the other hand, the presentation of the results would be significantly improved if they included a graph or table with some of the results. Numerical data is referred to throughout the text, which is always easier, clearer and more attractive to look at if presented graphically.

I hope that these suggestions help to improve the manuscript and that you finally get to see it published in this journal.

Kind Regards

Author Response

The authors provided a cover letter to explain the details of the revision in the manuscript (ijerph-914600) to response the reviewers' comments.

Changes that were done:

Title:

- In title was decided one term: Health Education Performed by Health Professionals to Child: Parent’s Perspective”

- To avoid an article unspecific the term “child/young” was replaced by “child”, and the term “parents/family” was replaced by “parents” in whole paper.

Abstract:

- It was put the portuguese context (line 22).

- The year of the data collection was revised (line 23).

Keywords:

- It was selected other keywords, because the words used before were repeated in the title (line 32).

Introduction:

- It was inserted two chapters: “Health education concept” (line 47) and “Health education in Portugal” (line 57).

- To addresses the HE concept and the state of play the HE in Portugal it was considered four new sources in these two chapters (line 48-83 and line 470-483).

Materials and Methods:

- It was deepened the description of sampling and data collection process (the choice of inquiry technique (line 111); the number of questions (line 119); type of questions (line 119); the interview situation (line 131); place were interviews take place (line 149); the person who conducted semi-structured interviews (line 111); the chosen of interview technique (line 111); the socio-demographic characteristics of the interviewees (line 133); the corresponding literature of content analysis technique that was applied (line 138); provided information that the interviews were transcribed verbatim (line 145)).

- The year of the data collection was revised (line 132-133).

- It was clarified the relationship between the researcher and the study participants (line 155-156).

Results:

- The term “majority” was replaced for “mean” (line 164).

- It was clarified the option for using pre-established categories (line 170).

- The subsections 3.1 (“Perception of parents about the Health Education concept” - line 179); 3.2 (“Assessing the nurse’s practices in Health Education to child and parents” - line 192) and 3.3 (“Evaluation of parents about the Health Education practice” - line 252) were renamed.

- The interviews were better structured and it was used direct quotations in the interviews and used quotations marks.

- It was put “PH” for the interviews conducted in primary health care and “P” for those conducted in pediatric services.

- The words “speech” and “speeches” were replaced with the word “interview/interviews” in whole paper.

- In Table 1, it was eliminated “Source: own elaboration”.

Discussion:

- The statement in line 304 to line 310 was reformulated.

- The statement in line 373 to line 378 was reformulated.

- It was introduced the results of the Portuguese context (line 414).

- The statement “…in view of the result of this research, we can conclude, that nurses exercise educational practices are capable of effective social changes” was reformulated (line 422-423).

- It was inserted a chapter with the limitations of the study (Line 429)

Conclusions:

It was done recommendations for public health policy and practice (line 441-446).

Reviewer 2 Report

In introduction:

The introduction is based on references mainly in the Latin language, and we do not understand why. There is no evidence in the text.

Line 44 - The context and the however of the statement are not clear.

In Methods:

No reference is made to the method of analysis adopted.  It is necessary to contextualize the methodology used.

It would be interesting to publish, in an annex, the questions that were part of the data collection instrument.

Data collection took place in early 2017. Why does the ethical opinion date from 2018?

Line 72 - It is stated that the principal investigator was responsible for collecting the data.

The potential for the researcher to influence the study and for the potential of the research process itself to influence the researcher and her/his interpretations acknowledged and addressed? For example I question:

Is the relationship between the researcher and the study participants addressed? Does the researcher critically examine her/his own role and potential influence during data collection? Is it reported how the researcher responded to events that arose during the study?

In Results:

Line 106 - Suggested replacement of “majority” for “mean”.

By reading the methods, it is understood that they will use the deductive method to determine the categories. Then, in the results, they refer to pre-established categories (Line 112). There seems to be an incongruity here.

In Figure 1: add the caption to "HE" and eliminate: "Source: Own elaboration."

About the representation of participants and their voices.

Generally, reports should provide illustrations from the data to show the basis of their conclusions and to ensure that participants are represented in the report. The citations was illustrate but the categories should be quoted in "..." and should be integrated into the text. Or, add in Table 1 examples of quotes that illustrate each of the categories.

I would suggest this big improvement in this section.

In Discussion:

A little confused. 

In Conclusions:

Line 345: The efficacy and effectiveness of health interventions is assessed using other types of methodological approaches. A fact that cannot be inferred from here.

In References:

Bibliographic references are very limited and not comprehensive in the international literature on the subject.

The almost exclusive use of bibliographic citations from Latin journals with low impact should have an explicit reason for their use. I would suggest improving this section or justifying it.

In Funding:

What is found through a brief internet search, related to this project, are studies related to physical activity. Examples:

"Effects of Exercise on Biomarkers in Health and Disease: Some New Insights with Special Focus on Extreme Exercise and Healthy Ageing"

"The effects of dry-land strength training on competitive sprinter swimmers"

Is the reference to the "UID 04045/2020" project correct?

Author Response

(The authors gave the same response as above.)

Reviewer 3 Report

Review report

This paper deals with the topic of health education in Portugal and assesses on the basis of a cross-sectional empirical study the quality of health education performed by nurses to children or rather young people from the perspective of parents (N= 20).

In the following, please find my comments on your paper:

Title: The title is too unspecific. “child/young” and “parents/family perspective”: Please, decide for one term. Your empirical basis comprises assessments of 20 parents. I failed to understand why you chose the term “family perspective”.

Abstract: The Portuguese context is missing.

Keywords: They repeat the words used in the title. That is why it would be fine to select other keywords.

Introduction: There is a lack of context. The introduction chapter needs to be sharpened. What about the state of play of HE in Portugal? Why is this topic so relevant? Please, consider the following sources:

WHO Data on Portugal: https://www.euro.who.int/__data/assets/pdf_file/0007/355993/Health-Profile-Portugal-Eng.pdf?ua=1.

Early Childhood Intervention in Portugal: https://www.european-agency.org/file/14652/download?token=fVCHdKS1

https://www.younghealthprogrammeyhp.com/programmes/portugal.html

Some information on the HE Concept is provided in the introduction section. For HE is the key concept of this paper, a separate chapter is needed which addresses the HE Concept. This section should include information on the components of HE, the requirements of (successful) implementation and the pitfalls of implementation.

Materials and Methods: The empirical results presented in this paper are explorative-qualitative. That is why a precise description of the background of the study, the overarching aims of the study, the sampling and data collection process, the choice of inquiry technique, the questionnaire (number of questions, type of questions) as well as of the interview situation (Did you conduct interviews with one person each or did you talk with parents (“family”?)? Where did the interviews take place?) is needed. In the abstract you mentioned that you conducted semi-structured interviews. This information is missing in this section.

The duration of the interviews “was about 30 minutes” (line 79). With regard to the chosen interview technique (semi-structured interviews) this seems to me quite short. Please, explain.

What about the socio-demographic characteristics of the interviewees related to social status/educational background/marital status/income/family support?

What about the specific caregiving and nursing situations and the requirements related to the needs or specific situations of the children?

In line 84 the reader gets to know that you applied “the content analysis technique”. What kind of content analysis did you apply and why? The corresponding literature is missing.

Line 88: “Then, the data was theoretically framed …”. What do you mean? Please explain.

Line 90ff: Please add relevant literature and provide information on whether you transcribed the interviews verbatim or mutatis mutandis.

Results: I suggest to rename the titles of the subsection and to present the information in a more structured manner. For instance, relating subsection 3.2, in my opinion, to insert the word “Assessing the nurses’ practices” would be a good choice.

Due to the character of the empirical study it is fine to use direct quotations from the interviews.

The abbreviations F and G are confusing. I think “PH” for the interviews conducted in primary health care and “P” for those conducted in pediatric services would be more appropriate.

Please, highlight all direct quotations. The use of quotations marks could be a good choice.

The title of subsection 3.3. is too fuzzy.

Please, replace the words “speech” and “speeches” with the word “interview/interviews”. This applies for the whole paper.

Discussion: In my opinion, much information basing on the literature review provided in this chapter should be shifted to the new section that presents the HE Concept.

Please provide more information on the results found in “other studies” (line 214).

Line 281 to line 286: Please, insert sources.

The discussion of the results in the Portuguese context is missing (cf. line 323ff). In my opinion, the statement “…in view of the result of this research, we can conclude, that nurses exercise educational practices are capable of effective social changes.” (line 327f), is overshooting (N = 20!).

Conclusions: The content of this section needs to be sharpened. Please, add recommendations for public health policy and practice.

Additional suggestions:

A division of the content in separate subsections could raise clarity.

Moreover, I recommend to sum up the results and the discussion in one chapter called “Results and discussion”.

Author Response

(The authors gave the same response as above.)

Round 2

Reviewer 2 Report

I would like to thank the authors for their prompt reply.

I have only a few questions to ask.

In line 164:”mean between 33-34 years”. The mean is a measure of central tendency. It must be represented in a value and not in a range.

In line 170: “According the objectives of the study and theoretical framework of the study”. Could you please citation the theoretical model used.  This is an important aspect to understand the categorization used.

This statement conflicts with what is written between the lines 139 to 142: “organized in three different chronological moments: pre-analysis; exploration of the material and the treatment of the results: inference and interpretation. In pre-analysis, the “floating reading” of the interviews allowed to organize and systematize ideas, organize the material, know the context of the 141 information and highlight the generic guidelines.”

Was the deductive or inductive method used? The statements present in the results and methods are conflicting. It is essential that this issue is clear.

In line 190: “To nurses”. (P3) - This sentence is very ambiguous.

This aspect was not answered:

In line 457: “Funding: This research was funded by National Funds through FCT - Foundation for Science and Technology financed by the project UID04045/2020.”

What is found through a brief internet search, related to this project, are studies related to physical activity. Examples:

"Effects of Exercise on Biomarkers in Health and Disease: Some New Insights with Special Focus on Extreme Exercise and Healthy Ageing"

"The effects of dry-land strength training on competitive sprinter swimmers"

Is the reference to the "UID 04045/2020" project correct?

Author Response

Changes that were done:

Results:

  • The statement in line 164 was reformulated;
  • In line 109-121 it was clarified the theoretical model used;
  • In line 137 it was included a Figure with the questions used in the study, to be more clearer and attractive to look
  • In line 204 the sentence was reformulated.
  • In line 470 the reference to the "UID 04045/2020" project is correct.

Reviewer 3 Report

Dear Authors,

Thank you for the comprehensive revision of your paper.

You have taken into account all my comments and recommendations.

That is why in my opinion, the paper is now ready for publication.

All the best.

Author Response

(The authors gave the same response as above.)
